# An Engineered Nanocomplex with Photodynamic and Photothermal Synergistic Properties for Cancer Treatment

**DOI:** 10.3390/ijms23042286

**Published:** 2022-02-18

**Authors:** Eli Varon, Gaddi Blumrosen, Moshe Sinvani, Elina Haimov, Shlomi Polani, Michal Natan, Irit Shoval, Avi Jacob, Ayelet Atkins, David Zitoun, Orit Shefi

**Affiliations:** 1Faculty of Engineering, Bar-Ilan University, Ramat Gan 5290002, Israel; elivaron3@gmail.com (E.V.); gaddi.b@gmail.com (G.B.); sinvanm@gmail.com (M.S.); elinhaimov@gmail.com (E.H.); michaln1000@gmail.com (M.N.); 2Bar-Ilan Institute of Nanotechnology and Advanced Materials, Bar-Ilan University, Ramat Gan 5290002, Israel; shlomipol@hotmail.com (S.P.); ayelet.atkins@gmail.com (A.A.); david.zitoun@biu.ac.il (D.Z.); 3Department of Chemistry, Bar-Ilan University, Ramat Gan 5290002, Israel; 4Faculty of Life Sciences, Bar-Ilan University, Ramat Gan 5290002, Israel; irit.shoval@mail.huji.ac.il (I.S.); avi.jacob@biu.ac.il (A.J.); 5Gonda Multidisciplinary Brain Research Center, Bar-Ilan University, Ramat Gan 5290002, Israel

**Keywords:** engineering, biomaterials, phototherapy, cancer, nanoparticles

## Abstract

Photodynamic therapy (PDT) and photothermal therapy (PTT) are promising therapeutic methods for cancer treatment; however, as single modality therapies, either PDT or PTT is still limited in its success rate. A dual application of both PDT and PTT, in a combined protocol, has gained immense interest. In this study, gold nanoparticles (AuNPs) were conjugated with a PDT agent, meso-tetrahydroxyphenylchlorin (mTHPC) photosensitizer, designed as nanotherapeutic agents that can activate a dual photodynamic/photothermal therapy in SH-SY5Y human neuroblastoma cells. The AuNP-mTHPC complex is biocompatible, soluble, and photostable. PDT efficiency is high because of immediate reactive oxygen species (ROS) production upon mTHPC activation by the 650-nm laser, which decreased mitochondrial membrane potential (^∆^ψ_m_). Likewise, the AuNP-mTHPC complex is used as a photoabsorbing (PTA) agent for PTT, due to efficient plasmon absorption and excellent photothermal conversion characteristics of AuNPs under laser irradiation at 532 nm. Under the laser irradiation of a PDT/PTT combination, a twofold phototoxicity outcome follows, compared to PDT-only or PTT-only treatment. This indicates that PDT and PTT have synergistic effects together as a combined therapeutic method. Our study aimed at applying the AuNP-mTHPC approach as a potential treatment of cancer in the biomedical field.

## 1. Introduction

Targeted cancer therapy aims at delivering medicine to a preferential and designated site while reducing systemic dosage [1]. Today, the conventional drug distribution system bypasses delivering medicine at effective concentrations selectively to tumors and is associated with many side effects [2]. Recently, the use of nanoparticles (NPs) in medicine and particularly drug delivery has received immense interest [3,4,5,6]. Hence, NPs with measured dimension, surface charge, physicochemical properties, and modification of surface polymers allow improved biocompatibility and overall effective targeted delivery [7]. Particularly, near-infrared light-responsive NPs are favored because of their biological window (650–900 nm) that is easy to operate and has minimal scattering and absorbance of tissue for deep penetration [8].

Photodynamic therapy (PDT) involves the administration of a photosensitizer (PS) with local illumination parallel to the absorption peak of the administered drug [9,10]. Today, PDT can be targeted very precisely, has no long-term effect, is minimally invasive, and is a clinically accepted method for treating cancer patients [11,12,13]. The photochemical reaction is initiated by the absorption of a photon of light by a photosensitizer molecule in its ground state and stimulating it to an excited state [12]. Subsequently, the photosensitizer transfers energy to molecular oxygen (^3^O_2_), exciting it to its highly reactive singlet state (^1^O_2_) [14,15]. These singlet oxygen species produced are highly reactive, extremely toxic, and lead to cell damage [16]. PDT is conducted typically in the wavelength range between 600–800 nm [17]. This technique enables the light dose and treatment conditions to be moderated to prevent over- and undertreatment and can be repeated at the same site if necessary [18].

Meso-tetrahydroxyphenylchlorin (mTHPC) photosensitizer, known also as “Temoporfin” and by the proprietary name ‘Foscan^®^’, is one of the oldest second-generation PS drugs present and is one of the most widely studied drugs in PDT and photodiagnostics [19,20,21,22]. Clinical trials showed that PDT activated by mTHPC is safe and effective for treating cancer [23,24,25]. The mTHPC has many advantages over different PSs such as stronger phototoxicity, high triplet yield, and high absorption in the deep red (652 nm), which is highly beneficial for light penetration in tumor tissue [26,27,28]. Moreover, mTHPC has an extended lifetime in its triplet state that allows greater singlet oxygen production, improving the efficacy of PDT [29]. The mTHPC, like other anti-cancer drugs, amasses in malignant tissues to a far greater extent than in normal tissues [30,31,32]. Nevertheless, conjugated drugs to NPs demonstrate long circulation time and greater levels of drugs in tumor tissues compared to freely administered drugs due to the enhanced permeability and retention effect (EPR) [33].

On the other hand, photothermal therapy (PTT), also known as optical hyperthermia, relies on photothermal agents (PTAs) with high photothermal conversion for converting light into heat to destroy cancer cells [34]. Gold nanoparticles (AuNPs) have been considered efficient photothermal agents [35,36,37,38,39,40] due to their surface plasmon resonance (SPR) effect [41], which has a high efficiency of light-to-heat conversion [42]. When an electromagnetic field of light interacts with AuNPs, it causes a collective oscillation of electrons at the surface, also known as the localized surface plasmon resonance (LSPR), at a specific frequency [43]. AuNPs typically show the SPR band around 520 nm in the visible region [39,44,45]. At the biological molecular level, hyperthermia of PTT-mediated AuNPs induces cell membrane destruction, DNA denaturation, and ultimately cell death [46]. Overall, hypothermia depends on the light dose duration, drug dose, and intensity of the light [47,48].

In the present study, we synthesized a biocompatible AuNP–mTHPC complex and investigated its ability for a dual PDT and PTT treatment. Our hypothesis was that the amalgamation of PDT and PTT treatment increases efficacy compared to a mono-phototherapy approach, thus increasing overall cell death and consequently reducing the duration and intensity of laser irradiation needed. Therefore, we designed a photostable and active drug complex that can be activated by either a 532-nm or 650-nm laser. The 650-nm laser (PDT) activated the mTHPC conjugated drug and led to high ROS production, and the 532-nm laser (PTT) stimulated the AuNP, exerting a high photothermal conversion efficiency. The as-obtained AuNP–mTHPC complex showed enhanced synergistic anti-cancer effect in vitro by the combination of photodynamic and photothermal therapies in comparison to each therapy alone. Thus, our prepared AuNP–mTHPC may be applied as an effective dual-modal phototherapeutic agent for future clinical applications.

## 2. Results

Recently, we synthesized AuNPs and covalently conjugated mTHPC drug molecules through a 3-mercaptopropionc acid linker, as in our previous studies [22,49]. The characterization of the free mTHPC and AuNPs and in their conjugated complex (AuNPs−mTHPC) are described herein. Our conjugated complex was tested for its photothermal properties, photostability, ability to generate singlet oxygen molecules, and its toxic effect on the SH-SY5Y cell line as a potential tool for a combined PDT/PTT treatment of tumors.

### 2.1. Synthesis and Characterization of AuNP–mTHPC Complexes

The conjugation of AuNPs to mTHPC is explained in the Experimental section. We functionalized AuNPs with mercaptopropionic acid capping that served as a linker to conjugate the mTHPC. This conjugation occurs due to an esterification activity between the hydroxyl groups of mTHPC and the carboxylic group of the linker mercaptopropionic acid (Figure 1A).

The size of AuNPs and AuNP–mTHPC complex was determined through TEM measurements. The average diameter of the dry particles was 12.0 ± 0.5 nm for bare AuNPs and 12.0 ± 1.0 nm for AuNP–mTHPC (Figure 1B,C and Appendix A). The bare AuNPs were imaged at a higher concentration than the AuNP–mTHPC; Nevertheless the borderlines of each NP in both samples could be visualized distinctly. The average hydrodynamic diameter of each AuNP and AuNP−mTHPC solution was measured by DLS, exhibiting diameters of 52 ± 4 nm and 77 ± 8 nm, respectively (Figure 1D and Appendix A). This increase in size was due to the coating of mTHPC on the AuNP. The bare AuNP without functionalization exhibited a negative zeta potential of −7.4 mV. After conjugation to mTHPC, we observed the zeta potential to be −16.1 mV, indicating mTHPC attachment to the AuNP and successful conjugation (Figure 1E). The decrease in surface zeta potential after conjugation was due to the electronegative nature of the carboxylic groups of mTHPC [50].

An essential factor when conjugating active molecules to AuNPs is to retain their photoactivity when using light therapy. Hence, to verify the retention of photoactivity, the absorbance spectral properties of free mTHPC, AuNPs, AuNP–mTHPC, and irradiated AuNP–mTHPC were measured and are depicted in Figure 1F. The absorbance spectral properties of free mTHPC molecules are like those of the AuNP–mTHPC. Both spectra display a Soret band at 420 nm and four Q-bands, one of them a more prominent absorption band at a long wave of 650 nm [51]. The naked AuNP spectral properties display a peak at 532 nm, which is where we could direct a PTT laser. Afterwards, the AuNP–mTHPC complex was irradiated in the same manner used for the PDT/PTT treatment; the normalized absorbance did not change, attesting to the stability of this engineered complex and its future applications for repetitive PDT/PTT treatment.

### 2.2. Photothermal Imaging of AuNP–mTHPC Complexes

The resultant functional complex, AuNP–mTHPC, is a stable and soluble compound [22]. Herein, we were interested in using this complex for a combined PDT procedure with PTT. The AuNPs behaved as photothermal agents and mTHPC behaved as a photosensitizer [44,52,53].

The photothermal conversion of AuNP–mTHPC was examined by red (650 nm) and green (532 nm) lasers for 0–120 s and assessed by an infrared camera (Figure 2). We measured the temperature profiles of the irradiated complex using the same laser intensity of PDT and PTT during treatment. As estimated, the AuNP–mTHPC complex heating properties were strongly affected by the properties of the NPs, while the temperature of the control sample remained unaffected. When gold nanospheres were irradiated with 532 nm at 15 mW/cm^2^, under the condition used to stimulate the photothermal effect (PTT) of AuNP, the temperature increased by 0.8 °C after 120 s. However, under 650-nm irradiation at 6 mW/cm^2^, the condition used to induce the photodynamic effect (PDT) of mTHPC, the temperature rise was insignificant (Figure 2A), consistent with a prevalent photochemical reaction rather than a heating effect. The fluctuations during the infrared imaging were derived from the thermal noise, which was approximately 0.1 °C; the same fluctuations were seen in the pure water sample. As predicted, the heating profile of AuNPs was strongly affected by the 532-nm laser in comparison to the 650-nm laser due to the peak absorption spectrum of AuNPs [44,54].

### 2.3. Internalization of AuNP–mTHPC Complexes by SH-SY5Y Cells

Effective cellular uptake of NPs is one of the priority characteristics for this therapeutic, engineered AuNP–mTHPC complex [55]. To examine whether our AuNP–mTHPC was non-toxic, we incubated SH-SY5Y cells with AuNP–mTHPC for 24 h. We found that cells containing the AuNP–mTHPC complex grew similarly to the control cells and displayed no clear difference in cellular morphology (Figure 3A).

Next, we determined the kinetics of AuNP–mTHPC internalization in SH-SY5Y cells. Finding the time point for NPs’ uptake made our experiments more applicable and effective. We demonstrated AuNP–mTHPC cellular uptake using a high-throughput imaging flow cytometry method (ImageStreamX). We located a fluorescent signal after only 2 h (Figure 3B). However, after 18 h, the improvement in cellular uptake was incremental and showed similar results to 24 h (Figure 3C).

Thereafter, we sought to determine the specific localization of AuNP–mTHPC within the SH-SY5Y cells. Using high-resolution imaging with TEM-1400, we found that the AuNP–mTHPC complex could be found throughout the cytoplasm and particularly accumulated in numerous mitochondria (Figure 3D), in comparison to untreated cells (Appendix A). These different intracellular locations of the AuNP–mTHPC may cause parallel damage and trigger a more efficient PDT/PTT-induced cell death. Particularly mitochondrial photodamage can reduce ATP levels, inhibit the inner mitochondrial membrane enzymes, and eventually trigger a quicker apoptotic response [56,57,58,59]. This is primarily because a cell’s mitochondria are composed of sites where oxygen levels are excessive and, thus, ROS can be readily produced, initiating apoptosis [60].

We confirmed that the TEM images indeed contained the AuNP–mTHPC complex using a High-Resolution Scanning Electron Microscope-linked energy-dispersive X-ray (XHR-SEM-EDX) microanalysis unit. The EDX spectrum showed strong peaks of carbon and oxygen around 0.5 KeV for both untreated and treated AuNP–mTHPC cells. In addition, cells that were treated with AuNP–mTHPC had visible Au peaks in comparison to none for the untreated cells, confirming Au as a constituent (Appendix A).

### 2.4. The mTHPC Stability Properties and Reactive Oxygen Species’ (ROS) Generation Disturbs Mitochondrial Membrane Potential (^∆^ψ_m_)

Photostability is a central characteristic when tracking photosensitizers in vitro and in vivo. Furthermore, photostability is a crucial parameter for developing fluorescent photosensitizers because high photostability allows the phototherapy process to endure high-intensity laser scanning while lasting a long period with reduced photobleaching [61]. Continuous high-intensity bleaching by laser scanning microscopy was used to quantitatively measure the photostability of both the free mTHPC and AuNP–mTHPC complex.

As shown in Figure 4A, the signal loss was <20% for the AuNP–mTHPC complex during the initial 15 bleaches upon excitation at 561 nm. In contrast, more than 40% fluorescence signal loss was observed after the 15th bleaching iteration for the free mTHPC. Finally, by 105 bleaching iterations the signal intensity was more than double for the AuNP–mTHPC complex (25%) in comparison to the free mTHPC (10%). These results showed improved photostability for the AuNP–mTHPC complex relative to the free photosensitizer.

The intracellular ROS generation by AuNP–mTHPC in SH-SY5Y cells upon light irradiation was then assessed using 2′,7′-dichlorofuorescin diacetate (DCFDA) as an ROS indicator. Following incubation with AuNP–mTHPC and irradiation with a red light (6 mW/cm^2^) for 4 min, SH-SY5Y cells were examined using microscopy. As shown in Figure 4B, significant green fluorescence was found in SH-SY5Y cells upon irradiation. This was due to the DCFDA oxidation by the ROS produced from the AuNP–mTHPC, in comparison to particularly weak green fluorescence in cells without light irradiation. All the above results suggested that the AuNP–mTHPC is photostable in living SH-SY5Y cells and can be used for PDT because of excessive ROS productivity. As we know, singlet oxygen is highly reactive and damaging to membranes, particularly in the mitochondria [62,63,64]. Likewise, our AuNP–mTHPC complex concentrates in the cytoplasm and the mitochondria. Thus, we postulated that the increased production of singlet oxygen in mitochondria will cause damage to its membranes. To test this hypothesis, we measured the mitochondria membrane potential (^∆^ψ_m_) by studying the effect of the probe accumulation in mitochondria on its membrane potential using tetramethylrhodamine ethyl ester (TMRE) as an indicator [65].

TMRE is a fluorescent lipophilic cationic dye, which stains polarized mitochondria with high potential [62]. It is important to note that, although both AuNP–mTHPC and TMRE have red fluorescence, there was no interference signal obtained from the probe upon excitation at 560 nm.

In these experiments, SH-SY5Y cells were initially incubated with AuNP–mTHPC followed by PDT treatment. This was followed by co-staining with TMRE, and the confocal images are shown in Figure 4C. PDT-treated SH-SY5Y cells showed a decrease in mitochondrial membrane potential as measured by a tetramethylrhodamine ethyl ester (TMRE) fluorescent probe after 3 h. This suggested a disruption of mitochondrial function, whereas cyan fluorescence signal was observed for cells without light irradiation. These results indicated that irradiating the AuNP–mTHPC can depolarize the mitochondrial membrane potential and exert applicable cell death.

### 2.5. Phototoxicity of AuNP–mTHPC

Finding the laser power and duration parameters to determine an optimal cancer phototherapy was our main interest. The in vitro phototherapy effects of the PDT, PTT, and combined PDT/PTT were evaluated by using SH-SY5Y cells. The cells were incubated with AuNP–mTHPC for 24 h and then irradiated. For the PDT test, the cells were exposed to a 650-nm laser at 6 mW/cm^2^; for the PTT test, the cells were exposed to a 532-nm laser at 15 mW/cm^2^ (Figure 5A).

As shown in Figure 5B, the control conditions of either cells, laser, or AuNP–mTHPC alone had low cell death. Upon 650- and 532-nm light excitation for 4 min individually with the AuNP–mTHPC complex, 32.2 ± 7.5% and 32.6 ± 1.2% of cells were killed. However, when retaining the same time period of 4-min irradiation and dividing each treatment by half (2 min each), cell death increased to 46.9 ± 3.1% (Appendix A.). Nevertheless, when combining both treatments’ duration (4 min each), the cell death increased to 84.7 ± 5.5%, resulting in significantly lower cell viability compared to either PDT- or PTT-only treatment (Figure 5B). Thus, although the percentage of cell death was not considerably changed under single-laser irradiation (650 nm or 532 nm), the percentage of dead cells under laser irradiation at both wavelengths (650 nm and 532 nm) was significantly greater. These results distinctly demonstrated the combined consequences of PDT/PTT over any single modality treatment.

Furthermore, we analyzed cellular deformation after PDT/PTT since apoptotic cells usually appear shriveled, with condensed chromatin and fragmented nuclei [66,67,68]. As illustrated in Figure 5D, the ImageStreamX technique offers real-time visualization of AuNP–mTHPC distribution and cell shape for each single cell. As measured, cell shrinkage was identified in all light-treated cells (Figure 5E). Additionally, the index of circularity was highest for control condition cells, while irradiated cells showed loss of circularity. Still, PDT and PTT induced deformation, but less than the combined treatment (Figure 5F). Additionally, confocal microscopy was used to observe cellular damage and death (PI stain) following PDT/PTT. Consistent with flow cytometry analysis, we found greater differences between a combined PDT/PTT compared to PDT-only or PTT-only treatment, indicating visually improved success of the combined therapeutic method (Appendix A).

### 2.6. Dual PDT/PTT Treatment Results in a Synergetic Effect

Combining therapies for cancer treatment has become important [69]. Individually, each therapy relies on a mechanism-based reaction; yet, theoretically, combinatorial therapy might improve the total effectiveness of the therapy. Following the initial treatment, a portion of the cells will die, undergo damage, or remain unaffected. The second treatment affects both the damaged and unaffected cell populations.

The damaged cells induce statistical dependency between consecutive treatments (repetition of the same treatment, or a new one), as the damaged cells’ probability to either recover or die depends on the previous therapy efficiency probability. A combined treatment gain can be defined as the relative gain between two consecutive different treatments in comparison with the continuation of the first treatment (equal to repeating the same treatment twice) [70,71]. The combined treatment gain is:(1)$$S_{G}(n_{1},n_{2}|n_{1})=D_{P_{n_{1}},P_{n_{2}}}^{n_{1}+n_{2},T}-D_{P_{n_{1}},P_{n_{1}}}^{n_{1}+n_{1},T}$$
where $D_{P_{n_{1}},P_{n_{2}}}^{n_{1}+n_{2},T}$ and $D_{P_{n_{1}},P_{n_{1}}}^{n_{1}+n_{1},T}$ are the metastatic cell death probability in comparison to the reference (control groups) measured after T hours for treatment condition $n_{1}$, followed by treatment $n_{2}$, which can be either PDT or PTT in our setup. $P_{n_{1}}$ and $P_{n_{2}}$ are the treatment parameters for the first and second treatments, which, in our experiment setup, were the radiation power, I(mW/cm^2^), and treatment duration, τ (min).

When $S_{G}\left(n_{1},n_{2}|n_{1}\right)>0$, there is a synergistic effect where the two combined treatments have higher efficiency than having the same treatment repeated. When $S_{G}\left(n_{1},n_{2}|n_{1}\right)=0$, there is no synergetic gain, and the gain between treatments is additive. When $S_{G}\left(n_{1},n_{2}|n_{1}\right)<0,$ the gain efficiency is higher for the treatment $n_{1}$, and the gain will be antagonistic.

For unbiassed comparison of the combined effect and to compute the treatment gain, we equated the combined PDT and PTT laser powers and durations to individual PDT and PTT treatments. To detect and define synergy of sets of 4 and 8 min, we first reduced a baseline reference that included the average cell death of our control groups. The first set was PDT and PTT with an individual duration of 4 min, in comparison to a combination of PDT and PTT using 2 min of each laser. The second set included PDT and PTT durations of 8 min, in contrast to a combination of PDT and PTT using 4 min of each laser.

For the first set of 4-min duration, the cell death rate in relation to the controls was $D_{\tau_{PDT}=4}^{PDT,24}=29.5\pm 7.5$ and $D_{\tau_{PTT}=4}^{PTT,24}=29.9\pm 1.3$, and for the combined it was $D_{\tau_{PDT}=2,\tau_{PTT}=2}^{PDT+PTT,24}=44.2\pm 3.1$. By substituting the cell death rate in (1), we get the following combined treatment gains compared to the PDT and PTT, which separately were approximately $S_{G,4}\left(PDT,PTT|PDT\right)=14.7\pm 8.1$ and $S_{G,4}\left(PDT,PTT|PTT\right)=14.3\pm 3.3$, which signified a synergetic gain of 49% for PDT and 47% for PTT. For the second set of 8-min duration, the corresponding cell death rates were $D_{\tau_{PDT}=8}^{PDT,24}=66.4\pm 3.4$ and $D_{\tau_{PTT}=8}^{PTT,24}=81.1\pm 4.6$ and for the combined it was $D_{\tau_{PDT}=4,\tau_{PTT}=4}^{PDT+PTT,24}=82.1\pm 5$. For the longer treatment duration of 8 min, the gains were $S_{G,8}\left(PDT,PTT|PDT\right)=15.6\pm 6.5$ and $S_{G,8}\left(PDT,PTT|PTT\right)=0.9\pm 7.2$, which represented a synergetic gain of 23% for PDT and a negligible synergetic gain of 1% for PTT.

According to these results, we can tune our biologically based nanocomplex to have a synergistic effect. This confirmed a higher phototoxicity effect of the dual PDT/PTT treatment compared to PDT or PTT separately. We found that shorter treatment durations exhibited greater synergistic effect than longer treatment durations. Specifically, extending the laser duration for PTT led to saturation, where substantial cell damage was initiated and damaged cells were likely to die if either PDT or PTT was applied, i.e., the model became additive. Thus, this multimodal complex may be fine-tuned to cooperatively enhance interactions between mono phototherapies and induce a multimodal synergistic effect.

## 3. Discussion

To overcome the clinical limitations of single-modality therapies of PDT or PTT, we introduced a PDT/PTT dual-model therapeutic agent under laser irradiation. Our AuNP–mTHPC complex exhibited suitable biocompatibility and uptake by the cells. Remarkably, our complex was found concentrated in the cytoplasm and specifically in the mitochondria. Activation of oxidative stress by mTHPC–PDT was demonstrated by increased generation of ROS that led to a loss of mitochondrial membrane potential (^∆^ψ_m_). In addition, the AuNP−mTHPC complex endured its composition at high temperatures and was found to be more photostable than the free mTHPC drug. The particles exerted high photothermal conversion efficiency, which led to efficient photothermal therapy. Combining the irradiation conditions of PDT and PTT caused a synergistic reaction that resulted in the death of the entire cell population at low laser power and duration levels. Applying the AuNP–mTHPC as a multifunctional PDT/PTT material may overcome the current phototherapy limitations of high power, long duration, and overall non-efficient treatment. We offer a promising product that may be an effective treatment of solid malignant tumors. Future extension to this work can be the adaptation of the technique to in vivo experiments and then to clinical trials. In addition, since the system has many parameters that control the setup, we can create a data set that can optimize the cell death rate at a given laser power and duration. Thus, this work can be continued to personalized medicine where we can tailor the parameters of the PDT/PTT treatment to each patient in relation to his/her personal health condition.

## 4. Materials and Methods

### 4.1. Materials

Citric acid trisodium salt dehydrate, 99% pure, and 3-mercaptopropionic acid were purchased from ACROS ORGANICS. Hydrogen tetrachloroaurate(III) hydrate (99.9% Au) was purchased from System Chemicals. The meso-Tetrahydroxyphenylchlorin (mTHPC) was purchased from Frontier Scientific (Logan, UT, USA). Solvents such as EtOH and dimethylformamide (DMF) were purchased from Alfa Aesar (Ward Hill, MA, USA).

### 4.2. Preparation and Characterization of AuNP−mTHPC Conjugates

Synthesis of AuNPs was performed according to the Turkevich et al. method [69]. HAuCl4·3H2O (0.004 g) was dissolved in 20 mL of double-distilled water (DDW), stirred, and boiled. While still boiling, 2 mL of 38.8 mM sodium citrate was added (sodium citrate was used here as the reducing and capping agent). After about 10 min, the solution became deep red in color, and 5 μL of 3-mercaptopropionic acid was added for an additional 7 min. As the solution boiled, DDW was added as needed to keep the total volume of the solution near 22 mL. Although the heating of the final solution was stopped, stirring was maintained until room temperature was reached. This resulted in the formation of AuNPs functionalized with mercaptopropionic acid capping. The solution was then dialyzed for 24 h. The obtained AuNPs’ solution was used for the conjugation of mTHPC photosensitizer molecules to AuNP. This conjugation happened through an esterification procedure and included a preparation of 5 mg/mL mTHPC stock solution in ethanol. Next, the initial AuNPs’ solution was removed from the dialysis and the pH was tuned to 12. This basic solution was stirred and heated up to 80 °C for 150 min. When the solution reached 50 °C, some drops of the mTHPC solution (prepared earlier) were added at a constant rate. The pH during the procedure was maintained in the range of about 9–10. After heating, the solution was removed and dialyzed with DDW in basic pH for 3 days. The obtained solution was brown-red in color.

The structure and size of the nanoparticles were investigated using TEM (transmission electron microscope): JEOL 1400 at 120 kV, JEOL JEM-2100 (LaB6) at 200 kV, and Hitachi HF3300 for high- resolution transmission electron microscopy (HRTEM) imaging.

Dynamic light scattering (DLS) for hydrodynamic average diameter and zeta potential measurements were performed using a Zetasizer Nano-ZS (Malvern Instruments Ltd., Malvern, UK) employing a nominal 5-mW He−Ne laser (operating wavelength: 633 nm, 20 °C, triplicate measurements) and disposable DTS1060C-Cleare ξ cells (ddH_2_O, 25 °C). The absorbance values were characterized using ultraviolet-visible spectroscopy (UV-1650 PC; Shimadzu Corporation, Kyoto, Japan).

### 4.3. Cell Culture

Human neuroblastoma cells, SH-SY5Y (ATCC), were grown in tissue culture flasks (Greiner, Stroudwater, UK) containing high glucose Dulbecco’s modified Eagle’s medium (DMEM, Sigma-Aldrich, St. Louis, MO, USA). The medium was supplemented with 10% fetal bovine serum (FBS), 1% L-glutamine, 1% penicillin-streptomycin, and 0.2% amphotericin (Biological Industries, Beit-Haemek, Israel). The cells were incubated at 37 °C in a humidified atmosphere with 5% CO_2_/95% air and were subcultured twice a week by adding 5 mL of trypsin when reaching a confluence of ∼70%. The cell cultures were followed daily for up to a week using light microscopy (Leica DM IL LED; Leica Microsystems, Wetzlar, Germany).

These cells were used as a model to examine the nanoparticles’ uptake by human cancerous cells and cell viability. The 2′, 7′-Dichlorofluorescin diacetate (DCFDA) was used to measure intracellular ROS using the DCFDA Cellular ROS Detection Assay Kit (ab113851, Abcam) according to the manufacturer’s protocol. For mitochondria membrane potential detection, SH-SY5Y cells were incubated with 200 nM TMRE (ab113852, Abcam). Cells were grown on 13-mm glass coverslips, washed with PBS, and fixed for 20 min in 4% PFA. Nuclei were counterstained with Hoechst, and coverslips were mounted with mounting medium to prevent photobleaching.

### 4.4. TEM and HR-SEM

For the preparation of cellular TEM samples, SH-SY5Y cells were seeded and incubated on 10-mL culture plates at a density of 3 × 106 per plate for 24 h before the experiment. The cells were then treated with 1.2 μM of AuNP–mTHPC in serum-free media and co-incubated for another 24 h without light interference. The medium containing the AuNP–mTHPC nanoparticles was then discarded and the cells were completely washed with PBS three times. The cells were then fixed for 2 h in Karnovsky fixative (2.5% glutaraldehyde with 2.5% paraformaldehyde) in a 0.1 M sodium cacodylate buffer (pH 7.4) and washed with 0.1 M sodium cacodylate buffer. The cells were postfixed in 1% OsO4, 0.5% K 2 Cr2 O7, and 0.5% K 4 [Fe(CN)6] in 0.1 M cacodylate buffer (pH 7.4) for 1 h at room temperature, and then washed twice with 0.1 M cacodylate buffer followed by rinsing with DDW three times. Cells were then stained with 2% uranyl-acetate for 1 h, washed with DDW, dehydrated in ethanol, and embedded in Epon EMBED 812 (EMS). The resin was polymerized at 60 °C for 24 h. Ultrathin sections (70–90 nm) were obtained with a Leica ultracat (UC7). Ultramicrotome was then analyzed in a G-12 Spirit FEI electron microscope and a JEM-1400 JEOL electron microscope with an EDX attachment. Samples were inspected with a Magellan 400L (FEI–Teramo fisher) in STEM mode, which was equipped with an Oxford EDX detector. Images were taken at different resolutions.

### 4.5. Optical Characterization of AuNP–mTHPC

This experiment was designed to image the distribution of temperature over the sample area under laser illumination. The laser beam was directed at the sample from above. Two different lasers were used: one, a red diode laser at a wavelength of 650 nm, and the other, a green diode, pumped, solid-state laser at 532 nm. Temperature elevation over the sample was imaged using a radiometric thermal imaging camera (FLIR Systems Inc., Boston, MA, USA, model A325). The camera had 320 × 240 pixels and a temperature sensitivity of 0.07 °C. The spatial resolution of the camera was 0.5 mm. This kind of camera is sensitive to thermal radiation at a wavelength range of 8–14 μm and is completely blind to lasers and other light sources at the visible or near-infrared spectral ranges. For each experiment, a few seconds of ambient temperature were recorded (at the center of the laser beam). The irradiance was measured to be 6 mW/cm^2^ and 15 mW/cm^2^ at the center of the beam. To measure the photostability of the AuNP–mTHPC, cells were treated with 1.2 μM of free mTHPC or 1.2 μM of AuNP-mTHPC. Using a wide-field Leica DMI8 microscope (Mannheim, Germany) equipped with a laser scanning system, we utilized the LasX acquisition software’s scanner module as follows. First, four ROI with cells (free mTHPC *n* = 30, AuNP–mTHPC *n* = 30 cells) and one ROI with no cells for background, per image, were used. Then, the experiment was started whereby the cells were imaged and the ROIs were then bleached at maximum laser power for 15 iterations. This was automatically cycled six more times, providing multiple data points, with each point accumulating the bleaching iterations. Intensity data were exported to Excel, the background was subtracted, and the data were plotted.

### 4.6. Confocal and Wide-Field Microscopy

Confocal microscopy images were acquired using an inverted Leica DMi8 scanning confocal microscope, driven by the LASX software (Leica Microsystems, Mannheim, Germany). The objective used was CS2 63×/1.40 oil objective. Excitation for PI was with a 552 laser and emission was detected between 569 nm–640 nm, and for AuNP-mTHPC with a 630 laser and emission between 650–750. Wide-field images were acquired using a DMi8 Leica wide-field inverted microscope, equipped with an sCMOS DFC9000GT Leica camera using a 20 × 0.75 objective.

### 4.7. Flow Cytometry Studies

For imaging flow cytometry analysis, samples of SH-SY5Y cells were harvested, then centrifuged to obtain a pellet of about 105 cells in 30 μL. Images were acquired using an ImageStreamX MarkII (ISX, Amnis, Luminex) at 60× magnification. A total of 1000–2000 events per sample were acquired at minimum speed. The AuNP–mTHPC inside the cells was imaged using the APC channel (channel 11, ex641 nm, em640–745 nm). The resulting images were analyzed with the IDEAS 6.2 (AmnisCorp) software. To measure the kinetics of AuNP–mTHPC internalization in SH-SY5Y cells, the mean pixel intensity was calculated for each image. Next, cell size was measured using the area feature, and cell shape was measured using the circularity feature (whereby values close to zero indicated elongated/oblique cells, while high values indicated circular cells). These features were calculated over the object mask of the brightfield channel (BF) and compared between samples. Data are expressed as mean ± SE arbitrary fluorescence units of replicate samples, mean ± SE area, or mean ± SE circularity score ranking. To evaluate cell apoptosis induced by the PDT/PTT treatment of AuNP–mTHPC, we used the propidium iodide (PI) staining method followed by acquisition and analysis by flow cytometry, using a BD LSR Fortessa analyzer (BD Biosciences, San Jose, CA, USA). For combining PTT and PDT effects in vitro, SH-SY5Y cells were incubated in a 24-well culture plate at a density of 1 × 105 per well. The cells were seeded in DMEM (High Glucose) culture medium containing 10% fetal bovine serum (FBS, GIBCO) at 37 °C under a humidified atmosphere with 5% CO_2_. After 24-h incubation, the cells were treated with 1.2 μM of AuNP–mTHPC in serum-free media and co-incubated for another 24 h without light interference. Next, the cells were washed with DMEM to remove free NPs. Then, fresh medium was added, and the cells were exposed to a 650-nm laser (6 mW/cm^2^) for 2, 4, and 8 min (PDT irradiation), a 532-nm laser (15 mW/cm^2^) for 2, 4, and 8 min (PTT irradiation), or the combination of the PDT/PTT irradiation (2 or 4 min of each), respectively. After laser irradiation, the cells were incubated at 37 °C for 24 h. The cells were then collected, and PI was added, following the manufacturer’s recommendation. Samples were incubated in darkness for 5 min at room temperature and then analyzed using flow cytometry (BD LSRFortessa™). The acquired data were analyzed using FlowJo software (Ashland, OR, USA).

### 4.8. Modeling and Statistical Analysis

Data were processed and analyzed using MATLAB software (MATLAB version 2020.b). To examine the significancy between the control and the treated samples, we used a two-sample *t*-test. The results are presented as mean values ± STDEV or ±SE.

## Figures and Tables

**Figure 1 ijms-23-02286-f001:**
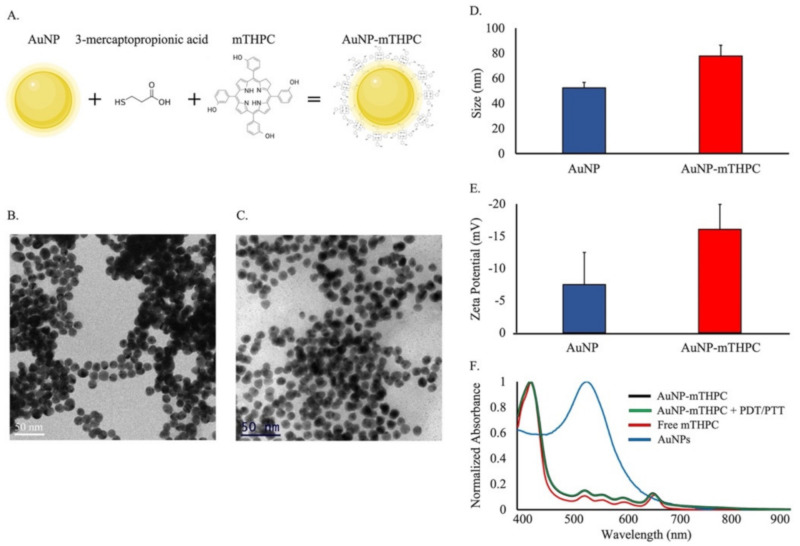
Size characterization and spectroscopic measurements of AuNP and AuNP–mTHPC. (**A**) AuNP–mTHPC complex illustration: a chemical structure of mTHPC in the conjugates and the linker 3-mercaptopropionc acid. (**B**) Left image depicts bare AuNPs, and (**C**) right image depicts AuNP–mTHPC. (**D**) Dynamic light scattering (DLS) measurement of the AuNP and AuNP–mTHPC. (**E**) Zeta potential measurements of AuNP and conjugated with mTHPC. (**F**) Normalized absorbance spectra of free AuNP (blue), AuNP–mTHPC (black), free mTHPC (red), and AuNP–mTHPC irradiated under 650-nm laser (PDT) at 6 mW/cm^2^ for 4 min and illuminated under 532-nm laser (PTT) at 15 mW/cm^2^ for 4 min.

**Figure 2 ijms-23-02286-f002:**
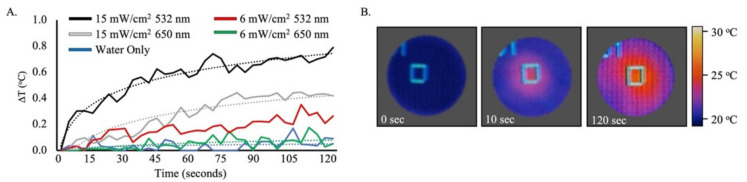
Heating properties of AuNP–mTHPC complex. (**A**) Temperature elevation of AuNP–mTHPC (0.1 mg/mL) as a function of irradiation time with the 650-nm (6 mW/cm^2^ and 15 mW/cm^2^) and 532-nm (6 mW/cm^2^ and 15 mW/cm^2^) lasers in an aqueous solution plotted with a trendline. (**B**) Examples of thermal images of AuNP–mTHPC monitored by an infrared camera at three time points (0, 10, and 120 s). The right scale represents the color code for surface temperature.

**Figure 3 ijms-23-02286-f003:**
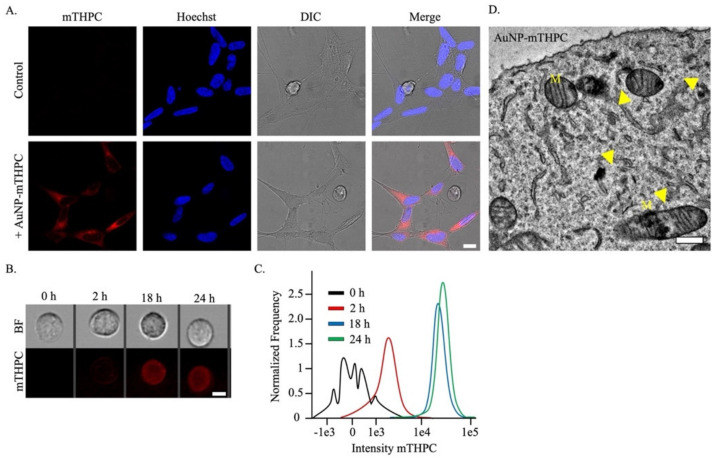
Localization of AuNP–mTHPC in live SH-SY5Y cells. (**A**) Images of SH-SY5Y cells incubated with 1.2 μm of the AuNP–mTHPC complex (red). Nuclei are counterstained with Hoechst (blue). Scale bar = 10 μm. (**B**) Images captured by ImageStreamX on the bright field (BF) and mTHPC (red) fluorescence channels, after 0, 2, 18, and 24 h of AuNP–mTHPC incubation. Scale bar = 7 μm. (**C**) Dynamics of AuNP–mTHPC entry into the cells measured by ImageStreamX. A histogram of the pixel-by-pixel intensity of mTHPC from SH-SY5Y cells (*n* = 1000) incubated with 1.2 μM of AuNP–mTHPC. (**D**) Electron micrograph of SH-SY5Y cell after incubation with 1.2 μm of the AuNP−mTHPC complex. The image shows the intracellular AuNP–mTHPC complex scattered through the cytoplasm and inside multiple mitochondria (M). Scale bar = 0.5 μm.

**Figure 4 ijms-23-02286-f004:**
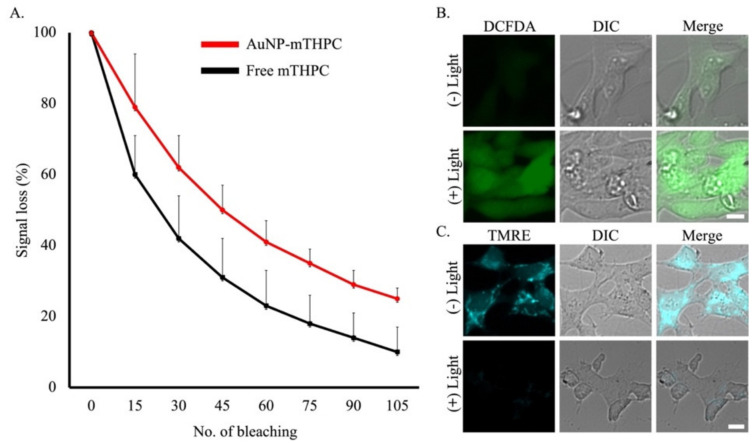
Photostability of AuNP–mTHPC and induced reactive oxygen species (ROS) that decreases mitochondrial membrane potential (^∆^ψ_m_) after PDT. (**A**) Signal loss (%) of fluorescent emission of free mTHPC and AuNP–mTHPC with the increasing number of bleaching iterations. Excitation wavelength: 561 nm for free mTHPC and AuNP–mTHPC. The average quantification of three repeated experiments is presented in the plot (mean ± STDEV). (**B**) Images of SH-SY5Y cells incubated with the ROS-reactive dye DCFH-DA to measure the level of ROS. Green fluorescence signal indicates the presence of DCF oxidized from DCFH. Detection of intracellular ROS production by DCFDA in SH-SY5Y cells after incubation with 1.2 μM of AuNP–mTHPC, with or without 650-nm light irradiation (6 mW/cm^2^, 4 min). Scale bar = 10 μm. (**C**) Images of SH-SY5Y cells incubated with the TMRE to evaluate mitochondrial membrane potential (^∆^ψ_m_). Cyan fluorescence signal indicates mitochondrial membrane activity. Detection of intracellular mitochondrial membrane activity by TMRE in SH-SY5Y cells after incubation with 1.2 μM of AuNP–mTHPC, with or without 650-nm light irradiation (6 mW/cm^2^, 4 min). Scale bar = 10 μm.

**Figure 5 ijms-23-02286-f005:**
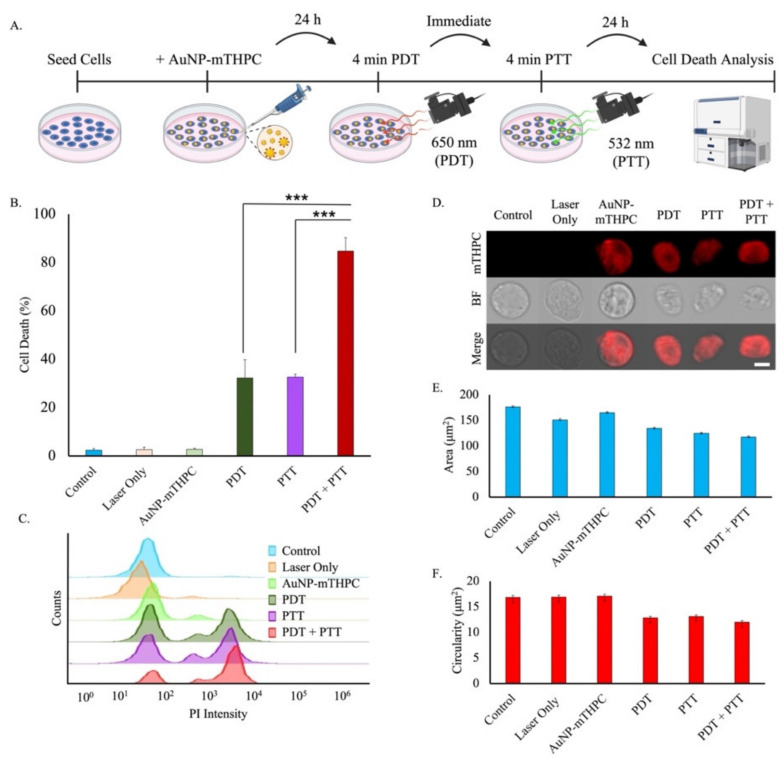
Cell death and cellular morphology changes induced by laser irradiation after 24-h incubation with AuNP–mTHPC complex. (**A**) Schematic representation of PDT/PTT procedure. (**B**) Cell viability of SH-SY5Y cells with AuNP–mTHPC and illuminated under 650-nm laser (PDT) at 6 mW/cm^2^ for 4 min or/and illuminated under 532-nm laser (PTT) at 15 mW/cm^2^ for 4 min. The average quantification of three experiments is presented in the plots (mean ± STDEV). There were significant differences in some of the experiments. *** *p* < 0.001. (**C**) Flow cytometry PI representative results. (**D**) Instances of images from different conditions captured by ImageStreamX on the Bright Field (DIC) channel and mTHPC (red) fluorescence channels. Scale bar = 7 μm. (**E**) Averaged area of cells for each type of irradiation (mean ± SE). (**F**) Averaged circularity of cells for each type of irradiation (mean ± SE).

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
