# Peer review of "An Engineered Nanocomplex with Photodynamic and Photothermal Synergistic Properties for Cancer Treatment"

_ijms, 2022, doi:10.3390/ijms23042286_

Round 1

Reviewer 1 Report

An Engineered Nanocomplex with Photodynamic and Photothermal Synergistic Properties for Cancer Treatment

Previous work has showed the synthesis of rationally designed new nanoplatform and novel nanomaterials working with the dual light source PDT and PTT used in clinical medical applications and cancer therapy. These nanoparticles have great potential especially for the treatment of melanoma or cancer types just below the skin. The manuscript is well written, I have some minor comments:

Abstract

Photodynamic therapy (PDT) and photothermal therapy (PTT) are promising therapeutic meth-ods for cancer treatment; however, as single modality therapies, either PDT or PTT are still lim-ited in their success rate. A dual application of both PDT and PTT, in a combined protocol, has gained immense interest. In this study, gold nanoparticles (AuNPs) are conjugated with a PDT agent, meso-tetrahydroxyphenylchlorin (mTHPC) photosensitizer, designed as nanotherapeutic agents that can activate a dual photodynamic/photothermal therapy. The AuNP-mTHPC com-plex is biocompatible, soluable, and photostable. PDT efficiency is high because of immediate re-active oxygen species (ROS) production upon mTHPC activation by the 650 nm laser which de-creased mitochondrial membrane potential (∆ψm). Likewise, the AuNP-mTHPC complex is used as a photoabsorbing (PTA) agent for PTT, due to efficient plasmon absorption and excellent pho-tothermal conversion characteristics of AuNPs under laser irradiation at 532 nm. Under the laser irradiation of a PDT/PTT combination, a twofold phototoxicity outcome follows, compared to PDT-only or PTT-only treatment. This indicates that PDT and PTT have synergistic effects to-gether as a combined therapeutic method. Hence, applying our AuNP-mTHPC may be a potential treatment of cancer in the biomedical field.

Comment: The aim/objective should be included. Also, the type of cancer to be treated should be elucidated.

Methods

4.2. Preparation and Characterization of AuNP−mTHPC Conjugates.

4.3. Cell Culture

4.4. TEM and HR-SEM

4.5. Photothermal imaging of AuNP-mTHPC.

4.6. Confocal and wide-field microscopy.

4.7. Flow Cytometry Studies

4.8. Measuring Photostability

4.9. Modeling and statistical analysis

Results

2.1 Synthesis and characterization of AuNP-mTHPC complexes

2.2 Photothermal Imaging of AuNP-mTHPC complexes

2.3 Internalization of AuNP-mTHPC complexes by SH-SY5Y cells

2.4 mTHPC stability properties and reactive oxygen species (ROS) generation disturbs mitochondrial membrane potential (Δψm)

2.5 Phototoxicity of AuNP-mTHPC

2.6 Dual Lasers Application upon AuNP-mTHPC Results in a Synergetic effect

Comments: Suggest to enhance the cohesiveness between Methods and Results section.

In conclusion

combining the irradiation conditions of PDT and PTT caused a synergistic reaction that resulted in the death of the entire cell population at low laser power and duration levels. Applying the AuNP-mTHPC as a multifunctional PDT/PTT material may overcome the current phototherapy limitations of high power, long duration, and overall, non-efficient treatment. We offer a promising product that may be effective treatment of solid malignant tumors. Future extension to this work can be adaptation of the technique to in vivo experiments, and then to clinical trials.

In addition, since the system has many parameters that control the setup, we can create a mathematical prediction model that can predict the cell death rate at a given laser power and duration. Thus, this work can be continued to personalized medicine where we can tailor the parameters of the PDT/PTT treatment to each patient in relation to his/her personal health condition.

Discussion

We offer a promising product that may be effective treatment of solid malignant tumors. Future extension to this work can be adaptation of the technique to in vivo experiments, and then to clinical trials. In addition, since the system has many parameters that control the setup, we can create a mathematical prediction model that can predict the cell death rate at a given laser power and duration. Thus, this work can be continued to personalized medicine where we can tailor the parameters of the PDT/PTT treatment to each patient in relation to his/her personal health condition.

Comments: the Conclusion and Discussion seemingly promoting a particular product, aren’t it?

Author Response

Comment #1: The aim/objective should be included. Also, the type of cancer to be treated should be elucidated.

Answer #1: We thank the reviewer; we refined the abstract and included the cell line experimented in the paper.

Comment #2: Suggest to enhance the cohesiveness between Methods and Results section.

Answer #2: We improved the unity between methods and results section and rephrased some subtitles.

Comment #3: The Conclusion and Discussion seemingly promoting a particular product, aren’t it?

Answer #3: Overall,  our goal is to improve a dual phototherapy treatment, not necessarily to promote a particular product. There are no indications for this in the text. In the conclusion we improved the description of the future to build a personalized data set.

Reviewer 2 Report

  • Please add the unit "nm" for the first measured diameter and provide histograms for all the diameter measurements ib the supplementary
  • Why there is a high level of agglomeration for bare gold nanoparticles?
  • For the irradiation unit, some of cmare not superscripted on page 4, please correct them.
  •  What is the explanation behind the fluctuations in Figure 2A?
  • Please correct the reference in line 323.
  •  

Author Response

Comment #1: Please add the unit "nm" for the first measured diameter and provide histograms for all the diameter measurements in the supplementary.

Answer #1: We thank the reviewer for highlighting this important issue. Firstly, we corrected the units and added the nm in the corrected places. In addition, we retrieved the measurements of the diameters of the nanoparticles and provided a histogram in the supplementary (as can be seen in Figure S1).

Comment #2: Why there is a high level of agglomeration for bare gold nanoparticles?

Answer #2: We thank the reviewer for raising this important point, based on your comment we improved the description of the images showing different concentrations of particles within the samples. In the gold nanoparticle sample, there was a higher level of concentration that’s why it looks denser; however, in both images single and separate nanoparticles borderlines can be seen. We added in the text this point of concentration.

Comment #3: For the irradiation unit, some of cm2 are not superscripted on page 4, please correct them.

Answer #3: We corrected accordingly.

Comment #4: What is the explanation behind the fluctuations in Figure 2A?

Answer #4:  We thanks the reviewer for this  comment, we added in the text of the paper the explanation for these fluctuations which are derived from the thermal noise of the camera which is approximately 0.1°C. The same noise can be seen in the pure water.

Comment #5: Please correct the reference in line 323.

Answer #5: We apologize for this mistake and corrected.